# Comparison of Early Imaging and Imaging 60 min Post-Injection after Forced Diuresis with Furosemide in the Assessment of Local Recurrence in Prostate Cancer Patients with Biochemical Recurrence Referred for 68Ga-PSMA-11 PET/CT

**DOI:** 10.3390/diagnostics11071191

**Published:** 2021-06-30

**Authors:** Steffen Bayerschmidt, Christian Uprimny, Alexander Stephan Kroiss, Josef Fritz, Bernhard Nilica, Hanna Svirydenka, Clemens Decristoforo, Elisabeth von Guggenberg, Wolfgang Horninger, Irene Johanna Virgolini

**Affiliations:** 1Department of Nuclear Medicine, Medical University Innsbruck, 6020 Innsbruck, Austria; christian.uprimny@tirol-kliniken.at (C.U.); alexander.kroiss@i-med.ac.at (A.S.K.); bernhard.nilica@i-med.ac.at (B.N.); hanna.svirydenka@i-med.ac.at (H.S.); clemens.decristoforo@i-med.ac.at (C.D.); elisabeth.von-guggenberg@i-med.ac.at (E.v.G.); irene.virgolini@i-med.ac.at (I.J.V.); 2Department of Medical Statistics, Informatics and Health Economics, Medical University Innsbruck, 6020 Innsbruck, Austria; Josef.Fritz@i-med.ac.at; 3Department of Urology, Medical University Innsbruck, 6020 Innsbruck, Austria; Wolfgang.Horninger@i-med.ac.at

**Keywords:** prostate cancer, PET, PET/CT, prostate specific membrane antigen, 68Ga-PSMA-11, furosemide, early imaging, biochemical recurrence, local recurrence

## Abstract

Background: 68Ga-PSMA-11 PET/CT is a promising method for the assessment of local recurrence (LR) in prostate cancer (PCa) patients. The aim of this study was to evaluate the diagnostic performance of early 68Ga-PSMA-11 PET imaging in comparison to 68Ga-PSMA-11 PET imaging 60 min post-injection (p.i.) in the detection of LR in patients with biochemical recurrence (BR) of prostate carcinoma. Materials and Methods: 190 image sets of patients with BR in PCa who underwent 68Ga-PSMA-11 PET/CT were assessed retrospectively (median prostate specific antigen (PSA) value, 0.70 ng/mL (range, 0.1–105.6 ng/mL)). Patients received an early static scan of the pelvic area (median, 248 s p.i. (range, 56–923 s)) and a whole-body scan 60 min p.i. (median, 64 min p.i. (range, 45–100 min)) with intravenous administration of 20 mg furosemide i.v. at the time of tracer application, followed by intravenous hydration with 500 mL of sodium chloride (NaCl 0.9%). Assessment was based on visual analysis and calculation of the maximum standardized uptake value (SUVmax) of the pathologic lesions present in the prostate fossa found in the early PET imaging and 60 min PET scans. The scans were characterized as negative, positive, or equivocal. The results were compared, and the combination of early and 60 min p.i. imaging was evaluated. Results: Image assessment resulted in 30 (15.8%) positive, 17 (8.9%) equivocal, and 143 (75.3%) negative findings in early scans, and 28 (14.7%) positive, 25 (13.2%) equivocal, and 137 (72.1%) negative findings of LR in 60 min p.i. images. For combined image analysis, 33 (17.4%) cases were positive and 20 (10.5%) were equivocal. There was no statistical significance between the number of positive (*p* = 0.815), negative (*p* = 0.327), and equivocal (*p* = 0.152) findings. Furthermore, the combination of both scans showed no statistically significant differences for the positive and negative findings (*p* = 0.063). The median SUVmax was 4.9 (range, 2.0–55.2) for positive lesions in the early scans and 8.0 (range, 2.1–139.9) in the scans 60 min p.i. The median SUVmax for bladder activity was 2.5 (range, 0.9–12.2) in the early scans and 8.2 (range, 1.8–27.6) in the scans 60 min p.i. Conclusion: Early static imaging additional to 68Ga-PSMA-11 PET images acquired 60 min p.i. has limited value in patients prepared with furosemide and hydration, and showed no statistically significant change in the detection rate (DR) of LR and the number of equivocal findings. Based on our results, in departments following a protocol with forced diuresis, including furosemide, additional early static imaging cannot be routinely recommended for the assessment of BR in PCa patients.

## 1. Introduction

Prostate cancer (PCa) has the second highest cancer incidence in men worldwide and the fifth highest mortality rate [1]. Relapse following primary therapy is a common problem in high-risk PCa. Over recent years, positron emission tomography/computed tomography (PET/CT) imaging using radiolabeled prostate-specific membrane antigen (PSMA) ligands has been proven to be a useful diagnostic modality in patients with biochemical recurrence (BR) for the detection of local recurrence (LR) of PCa after primary therapy. Radiotracers using PSMA ligands have been proven to have high sensitivity and specificity in the assessment of BR even in patients with low PSA levels [2,3].

68Ga-labeled Glu-NH-CO-NH-Lys(Ahx)-HBED-CC (68Ga-PSMA-11) is the most commonly used PSMA ligand worldwide [4]. Image acquisition is usually conducted 60 min post-injection (p.i.) [5]. However, at this time, due to renal excretion, 68Ga-PSMA-11 accumulates in the urinary tract, especially in the urinary bladder. This might lead to missed LR lesions in patients with BR [6]. Furthermore, high tracer accumulation can cause a so-called halo artifact, a phenomenon that leads to an extinction of the PET signal surrounding areas with intense tracer uptake, such as the urinary bladder [7].

The joint EANMM and SNMMI procedure guidelines for PSMA ligand PET/CT recommend the administration of 20 mg of furosemide combined with oral hydration of 500 mL of water to reduce tracer accumulation in the urinary tract [8]. Recent studies of our group showed promising results for the use of furosemide at the time of tracer injection followed by an infusion of 500 mL of sodium chloride (NaCl 0.9%). The presented protocol demonstrated a significant reduction in tracer activity in the bladder, which led to a significantly higher detection rate (DR) of LR. Moreover, forced diuresis significantly reduced the number of bladder halo artefacts, while having no negative impact on physiological tracer accumulation [7,9].

Other approaches to enhance the diagnostic certainty in the detection of LR in 68Ga-PSMA-11 PET imaging have been described. Early 68Ga-PSMA-11 PET imaging of the pelvic area right after tracer injection, additional to whole-body imaging conducted 60 min. p.i., has been investigated and reported as helpful in the evaluation of possible malignant lesions. Due to absence of tracer activity in the bladder during the first minutes, the combination of early and 60 min p.i. 68Ga-PSMA-11 PET imaging significantly increased the DR of LR in PCa patients with BR [10,11].

While these two preparation and imaging techniques have yielded higher diagnostic certainty, the combination of both has not yet been evaluated. This study retrospectively compared early static PET imaging and PET scans 60 min p.i. after forced diuresis with furosemide for the assessment of LR in PCa patients with BR referred for 68Ga-PSMA-11 PET/CT.

## 2. Material and Methods

### 2.1. Patient Population

For this retrospective analysis, a total of 190 PCa patients who were referred for 68Ga-PSMA-11 PET/CT between 3 January 2018 and 28 February 2020 for assessment of BR after definitive primary therapy were extracted from our database. Analysis included PET/CT exams of patients partly evaluated in previous publications by our group [7,9]. The median PSA value at the time of image acquisition was 0.7 ng/mL (range, 0.1–105.6 ng/mL), the median Gleason score was 7 (range, 5–10), and the median age was 71 years (range, 44–87 years). 

One hundred and sixty-six patients (87.4%) were treated with primary radical prostatectomy (RPE), while 24 patients (12.6%) were treated with primary radiation therapy (RT). After RPE, 58 patients received further salvage RT, following androgen deprivation therapy (ADT) in 24 patients. Nine patients were treated with additional ADT. After primary RT, eight patients received ADT. The patients’ characteristics are presented in Table 1.

The study concept was presented to our institutional ethics committee. As the study was designed retrospectively, using data obtained for clinical purposes, formal ethical approval was not deemed necessary by the ethics committee, meeting the legal requirements of our country. Written informed consent was obtained from all patients prior to the examination. All procedures performed in this study were in accordance with the principles of the 1964 Declaration of Helsinki and its subsequent amendments [12].

### 2.2. Radiopharmaceutical

PSMA-11 (Glu-NH-CO-NH-Lys(Ahx)-HBED-CC; HBED = N,N’-bis [2-hydroxy-5-(carboxyethyl)benzyl]ethylenediamine-N, N’-diacetic acid) was obtained from ABX advanced biochemical compounds (Radeberg, Germany) in good manufacturing practice quality. For 68Ga-PSMA-11 preparation, a 68Ge/68Ga generator (IGG100; 1850 MBq reference activity) and an automated synthesis module by Eckert & Ziegler (Modular-Lab PharmTracer; Eckert & Ziegler, Berlin) was used in the same manner as described in previous studies [10,13]. The radiochemical purity of the final product was >92%, as analyzed by reversed-phase HPLC analysis. 

### 2.3. Imaging Protocol

68Ga-PSMA-11 PET/CT imaging was conducted using a dedicated PET/CT system in time-of-flight mode (Discovery MI and Discovery 690; GE Healthcare, Milwaukee, WI, USA). Patients received a median activity of 153.8 MBq (range, 119.3–185.3 MBq). 

All patients received 20 mg of furosemide intravenously (i.v.) at time of tracer administration.

Early static images of the pelvic area (one bed position with an axial field of view of 15.6 cm) were acquired right after tracer injection (median time, 259 s p.i. (range, 56–923 s)) with a duration of 2 min. For attenuation correction and anatomic localization, a low-dose CT of the same region was carried out before PET acquisition. After early imaging, the patients were hydrated with 500 mL of sodium chloride (NaCl 0.9%) i.v. The scans performed 60 min p.i. consisted of a whole-body PET scan (skull vertex to the upper thighs) in the three-dimensional mode (emission time: 2 min per bed position with an axial field-of-view of 15.6 cm per bed position) 60 min after tracer injection (median uptake time, 64 min p.i. (range 45–100 min)). One hundred and eight patients (56.8%) received a diagnostic contrast-enhanced CT scan. The contrast-enhanced CT scan parameters using “GE smart mA dose modulation” were: 100 or 120 kVp, 80–450 mA, Noise Index of 24, 0.8 s per tube rotation, slice thickness of 3.75 mm, and pitch of 0.984. A CT scan of the thorax, abdomen, and pelvis (shallow breathing) was acquired 40–70 s after injection of contrast agent (60–120 mL of Iomeron 400 mg/L, depending on patient’s body weight), followed by a CT scan of the thorax in deep inhalation. In the remaining 82 patients (43.2%) a low-dose CT scan was performed for attenuation correction of the PET emission data. Low-dose CT was also used for anatomical allocation of lesions with increased uptake found on the PET scans. The low-dose CT scan parameters using “GE smart mA dose modulation” were: 100 kVp, 15–150 mA, Noise Index of 60, 0.8 s per tube rotation, slice thickness of 3.75 mm, and pitch of 1.375. Reconstruction was performed with an ordered subset expectation maximization algorithm (OSEM) with four iterations per eight subsets. Images were corrected for random coincidences and scatter.

### 2.4. Image Analysis

All 68Ga-PSMA-11 PET/CT images were analyzed with dedicated commercially available software (GE Advance Workstation SW Version AW4.5 02), which allowed the review of PET, CT, and fused imaging data in axial, coronal, and sagittal slices. The early and 60 min p.i. PET images were interpreted separately and independently by two board-approved nuclear medicine physicians, who were blinded to the clinical patient data and the results of other exams. In the case of patients already included in previous studies, the readers were not aware of the results of those analysis. For assessment of LR, the Prostate Cancer Molecular Imaging Standardized Evaluation (PROMISE) diagnostic criteria for 68Ga-PSMA-11 PET/CT reporting in PCa proposed by Eiber et al. and the consensus criteria for image interpretation defined by Fanti et al. were used as a reference [14,15]. For evaluation of early images, any focal uptake in the prostatic fossa higher than the background was rated as positive for LR. Patients were judged as either positive or negative for LR. Cases in which a clear distinction between urinary activity and a pathologic lesion was not possible were classified as equivocal.

In addition, we reevaluated and double-checked the findings 60 min p.i. according to the recently published EANM standardized reporting guidelines v1.0 for PSMA-PET [16]. Following the guidelines, images with a score of 4 and 5 were rated as positive findings; images with a score of 1 and 2 were rated as negative, whereas images with a score of 3 were rated as equivocal. Applying these new criteria, no change in the final diagnosis was observed.

In the case of disagreement between the two readers, the images were reevaluated, and a final diagnosis for the early and 60 min p.i. PET images was reached in consensus. Furthermore, cases negative in both images, cases positive or equivocal in the early images and negative in the 60 min p.i. images were rated as negative, cases positive in both images, negative or equivocal in the early images and positive in the 60 min p.i. images were rated positive, and cases equivocal in both images were rated equivocal.

The number of positive, negative, and equivocal scans for LR were evaluated separately for the early and 60 min p.i. PET images, as well as for the combination of early and 60 min p.i. images to reach a final diagnosis.

In addition, the intensity of tracer uptake in the bladder and in lesions judged as LR was measured, using the maximum and mean standardized uptake value (SUVmax and SUVmean). For SUV calculations, the volumes of interest (VOIs) were generated automatically by the software described above with a manually adapted isocontour threshold centered on the organs of interest.

### 2.5. Statistical Analysis

Baseline characteristics were analyzed descriptively. The number of positive, negative, and equivocal findings on the early scans vs. the 60 min p.i. scans and the scans 60 min p.i. vs. a combination of both scans were tabulated in cross-tables and compared with McNemar’s test for paired observations. A significance level of a = 0.05 (two-tailed) was applied for all *p*-values. Statistical analyses were performed using SPSS, version 26.0 (IBM Corp., Armonk, NY, USA).

## 3. Results

### 3.1. PET/CT Studies 60 min p.i.

Regarding the detection of LR, 28 cases (14.7%) were rated positive for LR on the PET images 60 min p.i. Moreover, 137 cases (72.1%) were judged as negative, and 25 cases (13.2%) as equivocal.

The median SUVmax of lesions positive for LR was 8.0 (range, 2.1–139.9), while the median SUVmax of urinary bladder activity 60 min p.i. in all patients was 8.2 (range, 1.8–48.7).

The overall DR for at least one pathologic finding consistent with recurrent PCa was 61.6%. Seventy-one patients (37.4%) showed PET-positive lymph node involvement, and positive findings suggestive for distant metastasis were detected in 35 patients (18.4%), with PSMA-avid skeletal metastases in 25 cases (13.2%) and non-skeletal distant metastases in 10 cases (5.3%).

### 3.2. Early PET/CT Studies

In the early images, 30 cases (15.8%) were detected that showed focal tracer accumulation highly suggestive of LR. There was no statistically significant difference of the DR of LR between the early and 60 min p.i. imaging studies (15.8% vs. 14.7%; *p* = 0.815).

The number of negative cases was 143 (75.3%) for the early PET images (75.3% vs. 72.1%; *p* = 0.327).

Seventeen cases (8.9%) were rated as equivocal, resulting in no statistically significant difference of equivocal findings compared to the 60 min p.i. scans (8.9% vs. 13.2%; *p* = 0.152).

The median SUVmax of lesions positive for LR was 4.9 (range, 2.0–55.2), while the median SUVmax of urinary bladder activity in the early PET studies in all patients was 2.5 (range, 0.8–8.1).

Out of the total 28 cases rated positive in the 60 min p.i. images, 23 cases were also visible in the early images with 20 cases (71.4%) positive and three cases (10.7%) equivocal in both images. Five cases (17.9%) positive in the 60 min p.i. images could not be detected in the early images.

Furthermore, out of the total 25 findings rated as equivocal in the 60 min p.i. scans, five (20%) were judged as positive for LR in the early scans, nine lesions (36%) were also rated as equivocal in the early scans, and 11 lesions (44%) could not be detected in the early scans.

Out of the 30 cases positive in the early images, five (10%) could not be seen in the 60 min p.i. images, and out of the 17 equivocal findings, five lesions (29.4%) had no correlation with the 60 min p.i. images.

Combined analysis of both studies to reach a final diagnosis yielded 137 negative cases (72.1%), 33 cases positive for LR (17.4%), and 20 equivocal findings (10.5%). As all cases negative in the 60 min p.i. scans were judged as negative in the combined analysis, there was no change in negative findings (72.1% vs. 72.1% *p* = 1). Differences in the DR of LR (17.4% vs. 14.7%) and equivocal findings (10.5% vs. 13.2%) for the combined image analysis compared to 68Ga-PSMA-11 PET/CT 60 min p.i. alone did not reach statistical significance (*p* = 0.063)

The findings and statistical analyses are presented in Table 2 and Table 3.

Regarding verification of suspicious lesions judged positive for LR on 68Ga-PSMA-11 PET/CT, no histologic confirmation could be achieved, as biopsy of lesions considered LR is not usually performed in this clinical setting at our institution. However, 20 of the 108 patients (18.5%) that received an additional diagnostic CT were judged positive for LR. Out of these 20 lesions detected in the diagnostic CT, 14 (70%) were judged positive on 68Ga-PSMA-11-PET. Five (25%) CT-positive LR lesions were either only visible in the early PET images or were judged as equivocal in the PET images. One lesion (5%) was missed completely in the PET images. In four patients, PET imaging resulted in positive lesions suspicious of LR that could not be seen in the diagnostic CT scans.

## 4. Discussion

Different approaches for optimal imaging strategies for patients referred for 68Ga-PSMA-11 PET/CT after primary radical treatment have been discussed in the recent literature, most of which have confirmed the value of multiphasic imaging [3,17,18,19,20].

A recent study of our group demonstrated a significantly higher DR of LR in patients receiving furosemide at the time of tracer injection compared to patients without preparation. Nevertheless, there was no significant change in equivocal findings [9].

This retrospective study aimed to compare the DR and diagnostic certainty of LR in 68Ga-PSMA-11 PET/CT imaging 60 min p.i. and in additional early static imaging of the pelvic area when using furosemide in patient preparation. We tried to address the question of whether early PET scans yield an additional value in the diagnosis of LR in comparison to the evaluation of PET scans 60 min p.i. after forced diuresis with furosemide.

In the present study, a statistically significant difference in the DR of LR and of equivocal findings between the early imaging and PET scans 60 min p.i. could not be demonstrated. Moreover, combined evaluation of the early and 60 min p.i. images did not significantly change the DR of LR and the number of equivocal findings compared to evaluation of images 60 min p.i. alone.

However, additional early static PET scans yielded a lower number of equivocal findings and improved characterization of equivocal findings in scans 60 min p.i., as demonstrated in Figure 1.

Uprimny et al. demonstrated the value of additional early static PET imaging of the pelvic area in 203 PCa patients with BR referred for 68Ga-PSMA-11 PET/CT [11]. The authors described a significantly higher DR of LR for a combined imaging protocol compared to a single whole-body scan 60 min p.i. (24.6% vs. 12.8%)

Additional early scans helped in lesion characterization and resulted in a significantly reduction in equivocal findings in the PET scans 60 min p.i. (15.8% vs. 4.5% of patients).

Kabasakal et al. [21] performed additional early static imaging of the pelvic area in 28 PCa patients referred for 68Ga-PSMA-11 PET/CT either for primary staging or assessment of BR. The authors described the possibility of better evaluation of the prostate fossa compared to whole-body scans 60 min p.i.

In a study presented by Beheshti et al. [3], the authors evaluated the performance of multiphasic 68Ga-PSMA-11 PET/CT in early recurrent PCa. Early dynamic images of the pelvic area and standard whole-body scans 60 min p.i. were conducted in 135 PCa patients. In cases of equivocal findings in the whole-body scans, delayed image acquisition 2–2.5 h p.i. limited to the pelvic area and abdomen/thorax was performed (*n* = 97). Only patients that received early, 60 min p.i., and delayed scans of the pelvic area were considered for comparative analysis (*n* = 81), resulting in no significant difference in the DR of LR between the different acquisition times (10 vs. 13 vs. 10).

Nevertheless, the authors described additional imaging as helpful for the detection and characterization of equivocal findings detected in whole-body 68Ga-PSMA-11 PET/CT 60 min p.i. However, analysis regarding equivocal findings for LR was not performed.

Uprimny et al. performed additional early dynamic PET images and 68Ga-PSMA-11 PET/CT 60 min p.i. [10] Early dynamic images were obtained in the first 8 min after tracer application. In a group of 64 PCa patients with BR, 13 patients were judged positive for LR in both early dynamic images and images 60 min p.i. Furthermore, in six patients judged negative or equivocal in images 60 min p.i., early dynamic images revealed PSMA-positive findings suspicious of LR, increasing the DR of LR from 20.3% to 29.7%, while not reaching statistical significance. In the early dynamic images, all detected lesions were positive within the first 3 min after tracer application, while tracer accumulation in the bladder became intrusive 5 min p.i. in some patients.

A possible problem of late imaging 2–2.5 h p.i. might be the moderate affinity of tumor lesions to PSMA and slow internalization, which might lead to a rapid tracer washout in tumor lesions before it binds to the PSMA receptors [22]. Furthermore, additional late images, depending on the specific protocol, often require a longer overall acquisition time. Despite good performance regarding lesion characterization, it might be difficult to implement these additional scans into routine clinical workflows, especially in centers with high throughput of PET exams and limited camera availability.

In the above-mentioned studies by Uprimny et al. Kabasakal et al. and Beheshti et al. the patients received no preparation before whole-body scans were performed 60 min p.i. To date, no publications are available that evaluate the effect of additional early static imaging compared to 68Ga-PSMA-11 PET/CT 60 min p.i. in patients prepared with hydration and forced diuresis with furosemide.

In our study, we performed patient preparation with hydration and furosemide as recommended by the joint EANM and SNMMI procedure guidelines for PCa imaging [8]. Nonetheless, high bladder activity in some patients was present 60 min p.i., despite forced diuresis, which has also been described in previous studies [9,23,24,25].

Hence, it is possible that despite administration of furosemide, lesions with lower tracer uptake adjacent to the urinary bladder are still judged as equivocal or might even be missed in PET scans 60 min p.i.

As furosemide was injected at the time of tracer injection, a low residual diuretic effect in some patients might have been present, caused by the short biological half-life of furosemide of 2 h, resulting in an increase in tracer accumulation in the urinary bladder in scans 60 min p.i. [26].

Therefore, it might be useful examining whether later furosemide administration than described in our protocol might yield a positive effect on lesion detection and characterization in whole-body 68Ga-PSMA-11 PET/CT imaging 60 min p.i.

A limitation of this study protocol was its retrospective data collection. Furthermore, no histopathological verification of PET-positive lesions was performed, as it is not part of the standard clinical work-up of PCa patients with BR at our institution. However, interpretation was performed by two experienced readers in a standardized way following published guidelines [14,15,16]. Therefore, we conclude that PSMA-positive local findings in the prostate fossa that were not verified by other imaging modalities or on follow-up can be regarded as true positives.

## 5. Conclusions

In PCa patients with BR referred for 68Ga-PSMA-11 PET/CT acquisition in early images can be helpful in some cases to clarify equivocal findings in the prostatic fossa on the PET scans 60 min p.i. However, the recommendation to perform early PET scans regularly seems unjustified, as no statistically significant increase in the DR of LR could be found compared to the PET scans 60 min p.i. after patient preparation with hydration and furosemide injection simultaneously with the radiotracer.

## Figures and Tables

**Figure 1 diagnostics-11-01191-f001:**
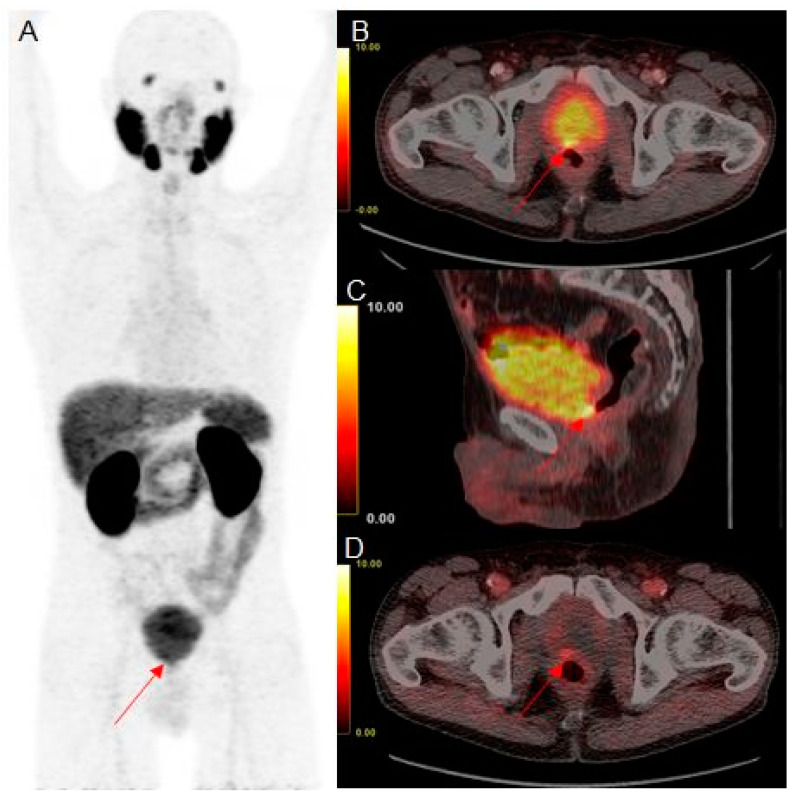
Example of an equivocal finding in the prostate fossa in 68Ga-PSMA-11 PET/CT scans 60 min p.i. with maximum intensity projection (**A**), fused axial (**B**), and fused sagittal (**C**) slices, and positive findings in the early fused axial (**D**) slices of a prostate cancer patient with biochemical recurrence after radical prostatectomy and salvage radiation therapy (PSA, 0.46 ng/mL). Intense focal uptake is the present paramedian at the posterior surface of the urinary bladder (red arrowhead). A clear distinction between local recurrence and urinary activity within the bladder is not possible in late images (SUVmax of focal uptake, 8.9; SUVmax of urinary bladder, 8.9). However, focal tracer accumulation is clearly visible in the early images (SUVmax of focal uptake, 3.3; SUVmax of urinary bladder, 2.7).

**Table 1 diagnostics-11-01191-t001:** Patient characteristics.

Patients	190
Primary radical prostatectomy	166
Primary radiation therapy	24
Salvage radiotherapy	58
Androgen deprivation therapy	41
Initial Gleason score, median (range)	7 (5–10) *
PSA (ng/mL), median (range)	0.7 (0.1–105.6)
Age (years), median (range)	71 (44–87)
Body mass index	26.1 (18.6–36.6)

* In 14 of 190 patients, no initial Gleason score could be obtained.

**Table 2 diagnostics-11-01191-t002:** Comparison of the early vs. 60 min p.i. scans.

	Early	60 min p.i.	*p*-Value *
Negative	143 (75.3%)	137 (72.1%)	0.327
Positive	30 (15.8%)	28 (14.7%)	0.815
Equivocal	17 (8.9%)	25 (13.2%)	0.152
SUVmax lesion	4.9 (2.0–55.2)	8.0 (2.1–139.9)	
SUVmax bladder	2.5 (0.9–12.2)	8.2 (1.8–48.7)	

* *p*-values from McNemar’s test for clustered binary paired data.

**Table 3 diagnostics-11-01191-t003:** Comparison of the 60 min p.i. vs. a combination of scans.

	60 min p.i.	Combination	*p*-Value *
Negative	137 (72.1%)	137 (72.1%)	1
Positive	28 (14.7%)	33 (17.4%)	0.063
Equivocal	25 (13.2%)	20 (10.5%)	0.063

* *p*-values from McNemar’s test for clustered binary paired data.

## Data Availability

The data presented in this study are available on request from the corresponding author.

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
