# Peer review of "Comparison of Early Imaging and Imaging 60 min Post-Injection after Forced Diuresis with Furosemide in the Assessment of Local Recurrence in Prostate Cancer Patients with Biochemical Recurrence Referred for 68Ga-PSMA-11 PET/CT"

_diagnostics, 2021, doi:10.3390/diagnostics11071191_

Round 1

Reviewer 1 Report

The Authors retrospectively evaluated the possible role of early acquisition of PET/CT with 68Ga-PSMA in the detection and localization of prostate cancer recurrence areas.

The topic appears to be extremely interesting in relation to the characteristics of the biodistribution of the tracer that involves elimination by urinary route, which may in fact make difficult the correct detection of lesions at the level of the prostate loggia.

However, the study does not seem to show a significant added value of early acquisition in the evaluation of patients with biochemical recurrence of prostatic disease.

The work, although well written and clear in its setting, shows some limitations: in particular, it is suggested to the authors to consider the radiation protection implications related to early PET/CT acquisition as contrary to the principle of justification and optimization of the European legislation on exposure to ionizing radiation. This issue is extremely important considering that the verification of inconclusive findings is performed with CT, further increasing the exposure for the patient.

In addition, the conclusions appear confusing because contradictory and therefore should be reformulated. giving a clear indication to the reader, if possible.

It is also necessary to modify:

(a) “Figure 1. Fused axial (B) ,  fused sagittal (C) slices”, should be reversed

  1. b) "A possible problem of delayed imaging might be moderate affinity of tumor lesion to PSMA and slow internalization, which might lead to a rapid tracer washout in tumor lesions". Please comment more extensively about this statement, in relation also to the results of other work reporting conflicting data.

  1. c) In the text there is no indication of the mode of acquisition of early images: please clarify this aspect.

  1. d) It would be desirable that the Authors mention the results of studies on the use of PET/CT images obtained with dynamic acquisitions.

Author Response

Reviewer Report 1

The work, although well written and clear in its setting, shows some limitations: in particular, it is suggested to the authors to consider the radiation protection implications related to early PET/CT acquisition as contrary to the principle of justification and optimization of the European legislation on exposure to ionizing radiation. This issue is extremely important considering that the verification of inconclusive findings is performed with CT, further increasing the exposure for the patient.

It has been proven that early PET imaging additional to 68Ga-PSMA-11-PET 60 min p.i. significantly increases detection rate of local recurrence (Uprimny et al. DOI: 10.1007/s00259-017-3743-z). Moreover, we could prove that patient preparation with 20 mg furosemide and hydration significantly increases detection rate of local recurrence in 68Ga-PSMA-11-PET 60 min p.i. (Uprimny et a. DOI: 10.2967/jnumed.120.261866). Furthermore, both protocols reduce the number of equivocal findings. The aim of our study was to evaluate whether additional early PET imaging has the same positive effect on detection rate in patients prepared with furosemide and hydration.

Concerning the exposure to ionizing radiation and effective dose for patients: All examinations are performed under the ALARA principal. The number of PET acquisitions does not result in a need for higher tracer activity. Additional low dose CT of the pelvic area has an effective dose lower than 1 mSv. As we conclude that additional early PET imaging of the pelvic area is not recommended as a standard procedure, we could reduce exposure to ionizing radiation for patients prepared with furosemide and hydration.

In addition, the conclusions appear confusing because contradictory and therefore should be reformulated. giving a clear indication to the reader, if possible.

We reformulated the abstract’s conclusion that they are concurrent.

It is also necessary to modify:

a) “Figure 1. Fused axial (B) fused sagittal (C) slices”, should be reversed.

Image order has been changed.

b) "A possible problem of delayed imaging might be moderate affinity of tumor lesion to PSMA and slow internalization, which might lead to a rapid tracer washout in tumor lesions". Please comment more extensively about this statement, in relation also to the results of other work reporting conflicting data.

The wording has been changed so the phrase is more coherent. This is an unconfirmed hypothesis and need further investigation in future studies.

c) In the text there is no indication of the mode of acquisition of early images: please clarify this aspect.

The imaging protocol for early static PET images has been added.

d) It would be desirable that the Authors mention the results of studies on the use of PET/CT images obtained with dynamic acquisitions.

There are currently few works available discussing additional early dynamic PET imaging with a large patient population. For the results of Beheshti et al. please refer to the discussion. Concurrent to our findings detection rate did not improve additional imaging. Nevertheless, that additional PET imaging was helpful in the assessment of equivocal findings.

The findings of our group on early dynamic imaging (DOI: 10.1007/s00259-016-3578-z), which was
one of the first studies to discuss this topic in a larger patient population, were added to the
discussion.

Reviewer 2 Report

In this manuscript the authors compared the performance of 68Ga-PSMA 11 PET/CT using early imaging and late imaging at 60 min p.i. and forced diuresis with i.v. hydration for detection of local recurrence in prostate cancer patients with biochemical recurrence. There was no significant difference in detection rate of LR or number of equivocal findings.

This is a very well written manuscript, which adresses an important question. The methodology is sound and the results clearly presented. A minor point would be that in the abstract I would rather conclude that based on these results and considering that an additional early imaging timepoint is time consuming and has logistic disadvantages it cannot be recommended as a routine procedure anymore especially for departments with only one scanner where time slots are urgently needed for other patients.

Reviewer 3 Report

The study is interesting and generally well-written. The observed low added value of the early acquisition phase does not represent a limitation of the manuscript per se. However, a few methodological concerns should be underlined.  

1) Table 1 indicates that the study sample is 190 patients. Among them, 166 received primary radical prostatectomy and 16 primary radiation therapy. What about the remaining 8 patients? Generally speaking, the authors did not provide a detailed description of enrolled patients. Table 1 should be implemented as much as possible. 

2) A subgroup of patients was submitted to ceCT (n=108), while 78 were studied uniquely using a low-dose CT. What about the remaining 4 patients? Moreover, it is not abundantly clear the reason for this choice as well as the added value of ceCT compared to low-dose CT. The eventual differences related to the used CT protocol are not commented on in the discussion. 

3) The authors interpreted PSMA PET images following the standardized images interpretation protocol published in 2017. However, the EANM standardized reporting guidelines v1.0 for PSMA-PET have been recently published (PMID: 33604691). It is not abundantly clear the reason for this methodological choice.  

Minor points: A number of inconsistencies should be checked: prostate cancer is abbreviated as PCa and PC, GMP is not spelled, line 58 Ref [4] should be moved before the punctuation. 

Author Response

Reviewer Report 3

1) Table 1 indicates that the study sample is 190 patients. Among them, 166 received primary radical prostatectomy and 16 primary radiation therapy. What about the remaining 8 patients? Generally speaking, the authors did not provide a detailed description of enrolled patients. Table 1 should be implemented as much as possible.

Thank you for this observation. Fortunately, it was a transmission error without effect on statistical analysis. All eight patients were treated with primary radiotherapy. We added patients’ age to table 1. All assessed patient characteristics are available for review. Furthermore, we implemented patients’ characteristics into the main text.

2) A subgroup of patients was submitted to ceCT (n=108), while 78 were studied uniquely using a low-dose CT. What about the remaining 4 patients? Moreover, it is not abundantly clear the reason for this choice as well as the added value of ceCT compared to low-dose CT. The eventual differences related to the used CT protocol are not commented on in the discussion.

Again, a transmission error no one had noted yet. None of the four missing patients was submitted to ceCT. Numbers and percentages were corrected.
At our institution patients are either referred for PET or PET in combination with a ceCT, depending on the diagnostic strategy of the referring physician.
As for the additional value of performed ceCT please refer to the corresponding paragraph in the results. The limited value of CT in the assessment of local recurrence in prostate cancer patients is well known and was not part of our study, as it focused on the value of additional early PET imaging.

3) The authors interpreted PSMA PET images following the standardized images interpretation protocol published in 2017. However, the EANM standardized reporting guidelines v1.0 for PSMA-PET have been recently published (PMID: 33604691). It is not abundantly clear the reason for this methodological choice.
This is a valuable point. At the time of image assessment these guidelines had not yet been published and we were following the recommendations by Eiber et al. (DOI: 10.2967/jnumed.117.198119) which are very similar to the recently published EANM guidelines.

Minor points: A number of inconsistencies should be checked: prostate cancer is abbreviated as PCa and PC, GMP is not spelled, line 58 Ref [4] should be moved before the punctuation.

We did another proofreading and changed some minor spelling mistakes including the mentioned abbreviations.

Round 2

Reviewer 1 Report

The authors have satisfactorily modified and implemented the paper, which in its current form is sufficiently clear and worthy of publication

Author Response

Thank you for your review.

Reviewer 3 Report

All the revisions made by the authors are reasonable with the exception of the following:

At the time of image assessment these guidelines had not yet been published and we were following the recommendations by Eiber et al. (DOI: 10.2967/jnumed.117.198119) which are very similar to the recently published EANM guidelines.

I understand the author's point of view. However, even though at the time of images analysis these guidelines had not yet been published, they are now freely available. In the reviewer’s point of view their use might significantly improve the methods of the study, making it up to date.  

Author Response

We added the EANM standardized reporting guidelines for PSMA-PET to our image analysis. PET images 60 min p.i. were reevaluated following the criteria proposed in the new guidelines, applying the 4-point scale for visual PSMA expression and the 5-point scale for interpretation of PSMA-PET findings. There was no change in detection rate and number of equivocal findings. 

Round 3

Reviewer 3 Report

Revisions made by the authors are reasonable.